# The role of CPAP as a potential bridge to invasive ventilation and as a ceiling-of-care for patients hospitalized with Covid-19—An observational study

Jonathan Walker[1]*, Shaman Dolly[1]◉, Liji Ng[1]◉, Melissa Prior-Ong[1]◉, Kalpana Sabapathy[2]

**1** Calderdale Royal Hospital, Calderdale and Huddersfield NHS Foundation Trust, Halifax, United Kingdom, **2** Department of Infectious Disease Epidemiology, The London School of Hygiene and Tropical Medicine, London, United Kingdom

◉ These authors contributed equally to this work.
* johnniewalker@doctors.org.uk

**Data Availability Statement:** Data cannot be shared publicly because of risk of re-identification of some patients of the database. Data are available for researchers who meet the criteria for access to

## Abstract

### Background

Continuous positive airway pressure (CPAP) ventilation may be used as a potential bridge to invasive mechanical ventilation (IMV), or as a ceiling-of-care for persistent hypoxaemia despite standard oxygen therapy, according to UK guidelines. We examined the association of mode of respiratory support and ceiling-of-care on mortality.

### Methods

We conducted a retrospective cohort analysis of routinely collected de-identified data of adults with nasal/throat SARs-CoV-2 swab-positive results, at the Calderdale and Huddersfield NHS Foundation Trust between 10th March-19th April 2020 (outcomes determined on 22nd May).

### Findings

Of 347 patients with SARs-CoV-2 swab-positive results, 294 (84.7%) patients admitted for Covid-19 were included in the study. Sixty-nine patients were trialled on CPAP, mostly delivered by face mask, either as an early ceiling of care instituted within 24 hours of admission (N = 19), or as a potential bridge to IMV (N = 44). Patients receiving a ceiling of care more than 24 hours after admission (N = 6) were excluded from the analysis. Two hundred and fifteen patients (73.1%) maximally received air/standard oxygen therapy, and 45 (15.3%) patients maximally received CPAP. Thirty-four patients (11.6%) required IMV, of which 24 had received prior CPAP. There were 138 patients with an early ceiling-of-care plan (pre-admission/within 24h). Overall, 103(35.0%) patients died and 191(65.0%) were alive at study end. Among all patients trialled on CPAP either as a potential bridge to IMV (N = 44) or as a ceiling-of-care (N = 19) mortality was 25% and 84%, respectively. Overall, there was

confidential data. The data underlying the results presented in this study are available from Tracy Owen (tracy.owen@cht.nhs.uk), Clinical Governance Facilitator, Quality and Safety Team, Calderdale and Huddersfield NHS Foundation Trust, Huddersfield Royal Infirmary, Acre Street, Lindley, Huddersfield HD3 3EA.

**Funding:** The senior author, Kalpana Sabapathy, received funding from the UK Medical Research Council (MRC) and the UK Department for International Development (DFID) under the MRC/DFID Concordat agreement; part of the EDCTP2 programme supported by the European Union. Grant Ref: MR/R010161/1 URL: https://mrc.ukri.org The funders had no role in study design, data collection and analysis, decision to publish, or preparation of the manuscript.

**Competing interests:** The authors have declared that no competing interests exist.

strong evidence for higher mortality among patients who required CPAP or IMV, compared to those who required only air/oxygen (aOR 5.24 95%CI: 1.38, 19.81 and aOR 46.47 95% CI: 7.52, 287.08, respectively; p<0.001), and among patients with early ceiling-of-care compared to those without a ceiling (aOR 41.81 95%CI: 8.28, 211.17; p<0.001). Among patients without a ceiling of care (N = 137), 10 patients required prompt intubation following failed oxygen therapy, but 44 patients received CPAP. CPAP failure, defined as death (N = 1) or intubation (N = 24), occurred in 57% (N = 25) of patients. But in total, 75% (N = 33) of those started on CPAP with no ceiling of care recovered to discharge—19 without the need for IMV, and 14 following IMV.

## Conclusion

Our data suggest that among patients with no ceiling-of-care, an initial trial of CPAP as a potential bridge to IMV offers a favourable therapeutic alternative to early intubation. In contrast, among patients with a ceiling-of care, CPAP seems to offer little additional survival benefit beyond oxygen therapy alone. Information on ceilings of respiratory support is vital to interpreting mortality from Covid-19.

## Strengths and limitations of this study

- Sample size relatively small.

- Study sample representative of hospitalised Covid-19 patients in UK.

- Previously unreported data on role of ceilings-of-care in hospitalised Covid-19 patients.

- Novel data on use of CPAP separated by indication.

## Background

SARS-CoV-2 infection which causes Covid-19 was declared a global pandemic by the World Health Organization (WHO) on March 11[th] 2020. An estimated seven million people have been infected globally, with approximately 400, 000 recorded deaths as of 11[th] June 2020 [1]. In the absence of definitive treatment for SARS-CoV-2, oxygen and respiratory support is the mainstay of management to prevent death. As such, information on the presence of ceilings-of-care which limit escalation of respiratory support is vital to interpreting mortality outcomes [2].

Disease severity from Covid-19 varies widely and invasive mechanical ventilation (IMV) has been used in 8–29% of hospitalised patients [3,4]. In the UK, data suggest that mortality in patients who require IMV is approximately 50% [5]. Current World Health Organization (WHO) guidelines for the management of Covid-19 recommend that a trial of continuous positive airway pressure (CPAP) ventilation may be considered in patients who remain hypoxemic despite standard oxygen therapy (via nasal prongs or Venturi face mask), however this was not initially the case earlier in the pandemic [6,7]. The use of CPAP in Covid-19 has been questioned [8], but in contrast to many other healthcare settings, CPAP is used in the UK in preference to High Flow Nasal Oxygen, largely due to concerns about oxygen supplies. In the UK, CPAP for the management of hypoxemic patients who are admitted to hospital with

Covid-19 is indicated as a potential bridge to IMV for patients with no ceilings to their potential care pathways, as a ceiling-of-care and to facilitate extubation [9]. This paper focuses on the first two of these three indications.

A ceiling-of-care is considered in order to avoid unwanted interventions that carry a high risk of failure and unnecessary suffering. All adults admitted to hospital are assessed for frailty and other factors which may make IMV on the Intensive Care Unit (ICU) inappropriate as a treatment modality [10]. When relevant, a treatment escalation and limitation plan is agreed by consensus between clinician, patient and family. This usually involves a "Do not Attempt Cardiopulmonary Resuscitation" order (DNACPR) which almost invariably excludes treatment with IMV. In this case, a trial of CPAP may be considered for patients who require greater respiratory support than standard oxygen therapy alone [10]. For patients unable to tolerate CPAP (for instance some patients find the facial attachment distressing), standard oxygen therapy may have to be used as the ceiling-of-care for respiratory support.

The mode of respiratory support used for patients with persistent hypoxemia despite standard oxygen therapy therefore involves a complex inter-play of disease severity, pre-morbid status, ceiling-of-care decisions, current local guidelines and local resources [9,10]. The aim of our study was to evaluate the outcomes of adult patients admitted to two hospital sites in the UK, who were treated according to national guidelines [9]. We examined factors associated with mortality, including mode of respiratory support and ceiling-of-care. We hope to shed light on the role of CPAP as potential bridge to IMV for patients with no ceilings to their potential care pathways, and as an escalation of respiratory support for patients with a ceiling-of-care.

## Methods

### Study setting, design and participants

The Calderdale and Huddersfield NHS Foundation Trust (CHFT) operates as a single centre, consisting of two acute hospitals on separate sites with over 800 hospital beds in total, and serving a population of 460,000 across Kirklees and Calderdale in West Yorkshire, England. Prior to the SARS-CoV-2 pandemic, there were approximately four high-dependency (HDU) beds across both hospital sites and ten ICU beds. The latter was increased to 26 ICU beds during the peak of Covid-19 admissions. We conducted a retrospective cohort analysis using routinely collected de-identified data of patients with nasal/throat swab positive results for SARs-CoV-2, admitted during the first six weeks of the Covid-19 epidemic in the region (between 10th March and 19th April). Outcome status (death, discharged alive or still an in-patient) was determined on 22nd May 2020.

### Patient and public involvement

Effective management of Covid-19 is a global priority. Patient and public were not involved in the design of the study as it involved a retrospective analysis of routinely collected data from the Electronic Patient Record (EPR). This data was fully anonymised and de-identified prior to being accessed for inclusion in the analysis and the study.

### Procedures

Electronic clinical records were examined, and data collection was adapted from the International Severe Acute Respiratory and Emerging Infection Consortium and WHO standardised case record proformas. Data was collated by the medical team within the respiratory department. In addition, we collected information on care home residency and ceiling-of-care

planning. We examined all adult patient data with SARs-CoV-2 swab positive results. Clinical management was according to the NHS Specialty specific guidelines [9]. These state that "CPAP is the primary mode of non-invasive respiratory support for hypoxic COVID19 patients. Suggested initial settings are 10 cmH2O + 60% oxygen".

In this study, patients not maintaining oxygen saturations over 92–94% on 40–60% oxygen via a Venturi mask were commenced on CPAP 10 cm H20 and 10 litres oxygen, adjusted according to physician discretion. CPAP on the ward was delivered by the Breas Medical NIPPY 3+© ventilator, with oxygen entrained from the wall via piped oxygen attached to a flow meter. Pressure could be adjusted as required, to a maximum of 15 cm H2O and flow could be adjusted to a maximum of 15 litres O2. The default interface used on the ward was a full face mask, but a total face mask was used in a small number of patients who could not tolerate this.

With only one exception, CPAP as a ceiling of care was started on the respiratory wards. CPAP as a bridge to IMV was started on the respiratory wards in most cases. These wards consisted of three bays of four beds, and four side-rooms consisting of one bed. All beds had access to a wall-mounted oxygen supply and could support the use of CPAP. One nurse would typically look after four to eight patients.

Oxygen requirements, which were administered as the minimum required to maintain target oxygen saturations within the range set by national guidelines, were documented as a proxy marker for hypoxemia and hence for severity of disease.

Fourteen patients requiring CPAP were commenced on CPAP Hoods instead of face masks and this was always delivered on ITU, either via the Hamilton-S1©, or the Hamilton-C3© Ventilator. With one exception, all of the CPAP Hood patients remained for full escalation.

## Exposure and outcome variables and analysis

The primary outcome was mortality and the main exposure of interest was the maximal respiratory support received, defined as 'air/oxygen' for patients who were not escalated to either CPAP or IMV; 'CPAP' for patients who were not escalated to IMV; 'IMV' for patients who were intubated and ventilated (including prior air/oxygen and/or CPAP). A secondary exposure of interest was presence of an early ceiling-of-care plan (defined as pre-existing plans in place prior to admission or within 24h of admission) (S1 Fig).

The following baseline exposure variables determined by history and clinical assessment by clinicians, were assessed as potential confounding factors for death: sex, age category, ethnicity, body mass index (BMI), care home residency, pre-existing co-morbidities and clinical features at admission. Age was categorized as younger than 70y, 70-79y, 80-89y and ≥90y to ensure adequate outcome events in each category.

Continuous variables are presented as medians and inter-quartile ranges (IQRs) and categorical variables as counts and percentages. Missing values were excluded. Logistic regression was performed to estimate odds ratios (ORs). Likelihood ratio testing (LRT) was done to assess statistical evidence of association. The multivariable model included sex as a confounding factor *a priori*. Variables which showed evidence of association ($p < 0.05$) with both mortality and with respiratory support were examined for inclusion in the final multivariable model using a backward stepwise approach. The final multivariable model included age category, sex, respiratory rate at admission, ceiling-of-care plans (pre-admission/within 24h) and maximal respiratory support received. Patients who had a ceiling-of-care instituted more than 24h after admission were omitted as reverse causality was possible, i.e. that a ceiling-of-care was introduced following failure to respond to treatment.

All analyses were performed using Stata™ version 16.0 for Windows (Stata-Corp, College Station, Texas). Data analysed was departmental, routinely collected and fully anonymized. The Calderdale and Huddersfield NHS Foundation Trust Research and Development office considered this project a service evaluation to establish a standard, and did not require further approval. The database is available on request to the corresponding author.

## Results

### Patient characteristics in study population overall

During the first six weeks of the Covid-19 epidemic in the Calderdale and Kirklees region of West Yorkshire (between 10th March and 19th April), 347 adult patients (≥18y) had SARs-CoV-2 positive nasal/throat swabs (S2 Fig). We excluded patients who were not admitted (N = 18) or were admitted for reasons other than Covid-19, namely those who were swabbed prior to discharge or during hospitalization following admission for another unrelated reason (N = 35). Two-hundred and ninety-four (84.7%) patients were included in the final analysis and 215 (73.1%) of them maximally received air/standard oxygen therapy, 45 (15.3%) received CPAP and 34 (11.6%) received IMV. Twenty-four of the IMV patients had previously failed on CPAP and required further escalation.

Ages ranged from 23-102y (median age 71y (IQR 59,82)) (Table 1). One-hundred and eighty-three (62.2%) patients were male. The majority of patients were White (N = 239, 81.3%), 55 (18.7%) were Asian, 7 (2.3%) Black and 11 (3.7%) patients classified as Other ethnicity. Over one-fifth of patients were care home residents (N = 65, 22.1%).

The most common pre-existing co-morbidities were hypertension (N = 109, 37.1%) and chronic cardiac disease (N = 90, 30.6%) (Table 1). Fifty-six (19.0%) patients had dementia. The most commonly self-reported symptoms were fever (N = 190, 64.6%), cough (N = 211, 71.8%) and shortness of breath (N = 181 61.6%) and the median duration since onset of symptoms was 7 days (IQR 3,10). Temperature measured at admission ranged from 27.8–41.4 degrees Celsius (median 36.8, IQR 36.4, 37.5) and 52 (17.7%) had a fever (>37.8 degrees Celsius). One 97y old lady was found on the floor at home and was hypothermic (27.8 degrees Celsius). Respiratory rate at admission ranged from 14–64 breaths/minute (median 23, IQR 20,28) and heart rates ranged from 21–173 beats/minute (median 92, IQR 78,106). The majority of chest radiographs revealed bilateral opacities in 204 (69.4%) and 37 (12.5%) patients had unilateral opacities.

One hundred and fifty-seven of all patients (53.4%) had a ceiling-of-care plan of which 138 (87.9%) were instituted early (63 pre-admission and 75 within 24h of admission). The remaining 19 patients with a ceiling-of-care had it instituted later during admission. The median age of early ceiling-of-care patients (N = 138) was 81y (IQR 74,87) compared to median age of 59y (IQR 52,68) among patients with no ceiling-of-care (N = 137). Among early ceiling-of-care patients, standard oxygen therapy was the ceiling of respiratory support planned for 84 (60.9%) patients, CPAP for 41 (29.7%) patients and it was not pre-specified for 13 patients (9.4%) (the latter all went on to maximally receive air/oxygen). Ninety-five percent (N = 62) of care home residents had a ceiling-of-care, with the majority (64.5% (N = 40)) already having a ceiling plan before admission.

### Characteristics of patients by maximal respiratory support received

We examined patient characteristics separately by maximal respiratory support received (Table 2). Age and sex were strongly associated with maximal respiratory support received. Two care home residents received CPAP, but all remaining 63 maximally received air/oxygen therapy. There was statistical evidence of association ($X^2$ test) with maximal respiratory

**Table 1. Patient characteristics overall.**

| Patients overall | | N (%) |
|---|---|---|
| **Socio-demographic characteristics** | | |
| **Sex** | Female | 111 (37.8) |
| | Male | 183 (62.2) |
| **Age (y)** | Median (IQR) | 71 (59,82) |
| | 20–49 | 32 (10.9) |
| | 50–59 | 46 (15.7) |
| | 60–69 | 54 (18.4) |
| | 70–79 | 75 (25.5) |
| | 80–89 | 66 (22.5) |
| | ≥ 90 | 21 (7.1) |
| **Ethnicity** | White | 239 (81.3) |
| | Asian | 55 (12.6) |
| | Black & other[1] | 18 (6.1) |
| **BMI (N = 212)** | Median (IQR) | 27 (23,32) |
| | 15–19.9 | 14 (6.6) |
| | 20–24.9 | 63 (29.7) |
| | 25–29.9 | 59 (27.8) |
| | ≥ 30 | 76 (35.9) |
| **Care home resident** | | 65 (22.1) |
| **Pre-existing comorbidities[2]** | | |
| **Hypertension** | | 109 (37.1) |
| **Chronic cardiac disease** | | 90 (30.6) |
| **Diabetes mellitus without complications** | | 56 (19.1) |
| **Diabetes mellitus with complications** | | 26 (8.9) |
| **Dementia** | | 56 (19.1) |
| **Chronic pulmonary disease** | | 49 (16.7) |
| **Chronic neurological disorder** | | 38 (12.9) |
| **Asthma** | | 34 (11.6) |
| **Chronic kidney disease** | | 28 (9.5) |
| **Rheumatological condition** | | 22 (7.5) |
| **Malignant neoplasm** | | 20 (6.8) |
| **Clinical features at admission[3]** | | |
| **Days since onset of symptoms (N = 253)** | Median (IQR) | 7 (3,10) |
| **Fever** | | 190 (67.1) |
| **Cough** | | 211 (73.5) |
| **Shortness of breath** | | 181 (62.9) |
| **Temperature (degrees Celsius)** | Median (IQR) | 36.8 (36.4,37.5) |
| **Respiratory rate (breaths per minute)** | Median (IQR) | 23 (20,28) |
| **Heart rate (beats per minute)** | Median (IQR) | 92 (78,106) |
| **CXR at admission** | Clear[4] | 53 (18.0) |
| | Unilateral opacities | 37 (12.6) |
| | Bilateral opacities | 204 (69.4) |
| **Management** | | |
| **Ceiling-of-care** | None | 137 (46.6) |
| | Pre-admission | 63 (40.1) |
| | Within 24 hours | 75 (47.8) |
| | Later during admission | 19 (12.1) |

(*Continued*)

**Table 1.** (Continued)

| Patients overall | | N (%) |
|---|---|---|
| **Maximal respiratory support** | Air/oxygen | 215 (73.1) |
| | CPAP | 45 (15.3) |
| | IMV | 34 (11.6) |

[1] Black N = 7 & Other N = 11;

[2] Co-morbidities with at least 5% prevalence in study population;

[3] Three mostly commonly reported symptoms are shown;

[4] Thirteen patients had non-Covid related changes.

support received and the following pre-morbidities chronic cardiac disease (p = 0.008), dementia (p<0.001) and chronic neurological disease (p = 0.03) indicating a greater prevalence in patients maximally treated with air or oxygen, than patients treated with CPAP or IMV. Having diabetes mellitus with complications was also different depending on maximal respiratory support (p = 0.009), with the higher prevalence among CPAP patients, while asthma (p = 0.02) was more common among IMV patients.

Duration since onset of symptoms was strongly associated with maximal respiratory support received (p<0.001). Air/oxygen patients had a shorter median duration of symptoms of 5 days compared with CPAP and IMV patients (both 7 days). There was statistical evidence across clinical characteristics to indicate that patients maximally requiring CPAP or IMV had more severe disease than air/oxygen patients, based on reported symptoms, vital signs and chest radiography.

Among all air/oxygen patients, 28.8% (N = 62) had a ceiling-of-care plan in place prior to admission and a further 26.5% (N = 57) had a plan developed in the first 24h of admission. Nineteen patients (42.2%) maximally received CPAP and had an early ceiling-of-care, and all but one were planned during the first 24h of admission. Overall, ninety-two patients received maximal respiratory support with air/oxygen as planned, while among 76 patients who had a ceiling-of-care plan which allowed for escalation to CPAP, 51 (67.1%) received air/oxygen only. Fourteen patients with ceilings-of-care did not have mode of planned maximal respiratory support recorded.

## Patient outcomes

Overall 103 (35.0%) patients died and 187 (63.6%) recovered and were discharged. Four patients (1.4%) remained in hospital at the end of the observation period (median duration–38 days (IQR 35,40)). The overall number of days of hospitalization ranged from 0 to 42 days (ten patients were discharged and one patient died on the same day of admission), with a median of 7 days (IQR 3,11). The mortality among patients maximally requiring and receiving air/oxygen was 33.0% (N = 71); CPAP was 46.7% (N = 21); IMV was 41.2% (N = 14) (Fig 1). There was no evidence to suggest a difference in median duration of hospitalization between air/oxygen and CPAP patients (6 days (IQR 3,10) and 7 days (IQR 5,11), respectively ($X^2$ p = 0.11)) but there was strong evidence for a difference between each of these groups and IMV (24 days (IQR 7,36) ($X^2$ p<0.001)).

## Mortality by respiratory support received and ceiling-of-care

Patients who were hypoxemic despite standard oxygen therapy, and who required escalation to CPAP, had the highest mortality overall, especially in the sub-group with a ceiling-of-care

**Table 2. Patient characteristics by maximal respiratory support received.**

| | | Air/oxygen[1] 215 (%) | CPAP[2] 45 (%) | IMV[3] 34 (%) | $X^2$ test p-value[4] |
|---|---|---|---|---|---|
| **Socio-demographic characteristics** | | | | | |
| **Age (y)** | Median (IQR) | 76 (63,85) | 67 (56,73) | 59 (51,67) | <0.001 |
| | 20–49 | 19 (8.9) | 5 (11.1) | 8 (23.5) | <0.001 |
| | 50–59 | 26 (12.1) | 10 (22.2) | 10 (29.4) | |
| | 60–69 | 32 (14.9) | 11 (24.4) | 11 (32.4) | |
| | 70–79 | 54 (25.1) | 16 (35.6) | 5 (14.7) | |
| | 80–89 | 64 (29.8) | 2 (4.4) | 0 (0.0) | |
| | ≥ 90 | 20 (9.3) | 1 (2.2) | 0 (0) | |
| **Sex** | Female | 94 (43.7) | 9 (20) | 8 (23.5) | 0.002 |
| | Male | 121 (56.3) | 36 (80) | 26 (76.5) | |
| **Ethnicity** | White | 179 (83.3) | 35 (77.8) | 25 (73.5) | 0.36 |
| | Asian | 22 (10.2) | 8 (17.8) | 7 (20.6) | |
| | Black, Other | 14 (6.5) | 2 (4.4) | 2 (5.9) | |
| **BMI** | Median (IQR) | 26.4 (22.3,31.1) | 29.8 (24.6,33.5) | 29.2 (26.8,32.9) | 0.02 |
| | 15–19.9 | 13 (8.4) | 1 (3.7) | 0 (0.0) | 0.14 |
| | 20–24.9 | 51 (32.9) | 7 (25.93) | 5 (16.7) | |
| | 25–29.9 | 41 (26.5) | 6 (22.2) | 12 (40.0) | |
| | ≥ 30 | 50 (32.3) | 13 (48.2) | 13 (43.3) | |
| | *Missing* | *60 (27.9)* | *18 (40.0)* | *4 (11.8)* | |
| **Care home resident** | | 63 (29.3) | 2 (4.4) | 0 (0.0) | <0.001 |
| **Pre-existing comorbidities** | | | | | |
| **Hypertension** | | 77 (35.8) | 20 (44.4) | 12 (35.3) | 0.54 |
| **Chronic cardiac disease** | | 75 (34.9) | 12 (26.7) | 3 (8.8) | 0.008 |
| **Diabetes mellitus without complications** | | 42 (19.5) | 8 (17.8) | 6 (17.7) | 0.94 |
| **Diabetes mellitus with complications** | | 13 (6.1) | 9 (20.0) | 4 (12.1) | 0.009 |
| **Dementia** | | 54 (25.1) | 2 (4.4) | 0 (0) | <0.001 |
| **Chronic pulmonary disease** | | 36 (16.7) | 9 (20.0) | 4 (11.8) | 0.62 |
| **Chronic neurological disease** | | 35 (16.3) | 2 (4.4) | 1 (2.9) | 0.02 |
| **Asthma** | | 24 (11.2) | 2 (4.4) | 8 (23.5) | 0.03 |
| **Chronic kidney disease** | | 23 (10.7) | 4 (8.9) | 1 (2.9) | 0.35 |
| **Rheumatological condition** | | 19 (8.8) | 3 (6.7) | 0 (0) | 0.19 |
| **Malignant neoplasm** | | 17 (7.9) | 3 (6.8) | 0 (0) | 0.24 |
| **Clinical features at admission** | | | | | |
| **Days since onset of symptoms** Median (IQR) | | 5 (2,9) | 7 (4,12) | 7 (6,10) | <0.001 |
| *Missing* | | *38 (17.7)* | *2 (4.4)* | *1 (2.9)* | |
| **Fever** | | 32 (61.4) | 30 (66.7) | 28 (82.6) | 0.03 |
| *Missing* | | *9 (4.2)* | *0 (0)* | *2 (5.9)* | |
| **Cough** | | 141 (65.6) | 39 (86.7) | 31 (91.2) | 0.002 |
| *Missing* | | *7 (3.3)* | *0 (0)* | *0 (0)* | |
| **Shortness of breath** | | 123 (57.2) | 33 (73.3) | 25 (73.5) | 0.05 |
| *Missing* | | *5 (2.3)* | *0 (0)* | *1 (2.9)* | |
| **Temperature** | Median (IQR) | 36.7 (36.3,37.3) | 37.2 (36.7,37.7) | 37.4 (36.5,38.2) | <0.001 |
| **Respiratory rate** | Median (IQR) | 22 (19, 26) | 26 (20, 29) | 28 (24, 32) | <0.001 |
| **Heart rate** | Median (IQR) | 90 (74, 106) | 97 (78, 108) | 98 (86, 110) | 0.04 |
| **CXR at admission** | Clear[5] | 51 (23.7) | 2 (4.4) | 0 (0.0) | <0.001 |
| | Unilateral opacities | 35 (16.3) | 1 (2.2) | 1 (2.9) | |
| | Bilateral opacities | 129 (60.0) | 42 (93.3) | 33 (97.1) | |

*(Continued)*

**Table 2.** (Continued)

|  |  | Air/oxygen[1] 215 (%) | CPAP[2] 45 (%) | IMV[3] 34 (%) | $X^2$ test p-value[4] |
|---|---|---|---|---|---|
| **Ceiling of care timing** | Never | 83 (38.6) | 20 (44.4) | 34 (100) | |
| | Pre-admission | 62 (28.8) | 1 (2.2) | - | |
| | Within 24 hours | 57 (26.5) | 18 (40.0) | - | |
| | Later during admission | 13 (6.1) | 6 (13.3) | - | |

[1]Air only (N = 55), oxygen via nasal prongs (N = 72), via face mask (N = 54), via non-rebreather mask (N = 34);

[2] Seven patients had CPAP with helmet and 38 had CPAP with face mask;

[3] Ten patients escalated directly to IMV from air/oxygen while 24 were trialled on CPAP before IMV;

[4]Kruskal-Wallis equality-of-populations rank test was used to compare median values.

(Fig 1). However, when patients without a ceiling-of-care are examined separately, mortality is highest among IMV patients (41.2% N = 14), and only one patient in each of the air/oxygen (1.2%) and CPAP (5.0%) sub-groups died.

When all patients who ever received CPAP (including those who went on to require IMV N = 69) are examined, 19 patients had an early ceiling-of-care plan, and 84.2% of these patients died. Among those without any ceiling-of-care (N = 44), 75% recovered and were discharged–43.1% (N = 19) without requiring IMV and 31.8% (N = 14) after subsequently receiving IMV (Fig 2).

### Factors associated with mortality

Older age is strongly associated with a higher risk of death from Covid-19 (Table 3). Sex, BMI and residence in a care home were crudely associated with mortality but after adjusting for age there was no statistical evidence of association. This was also the case for a number of pre-

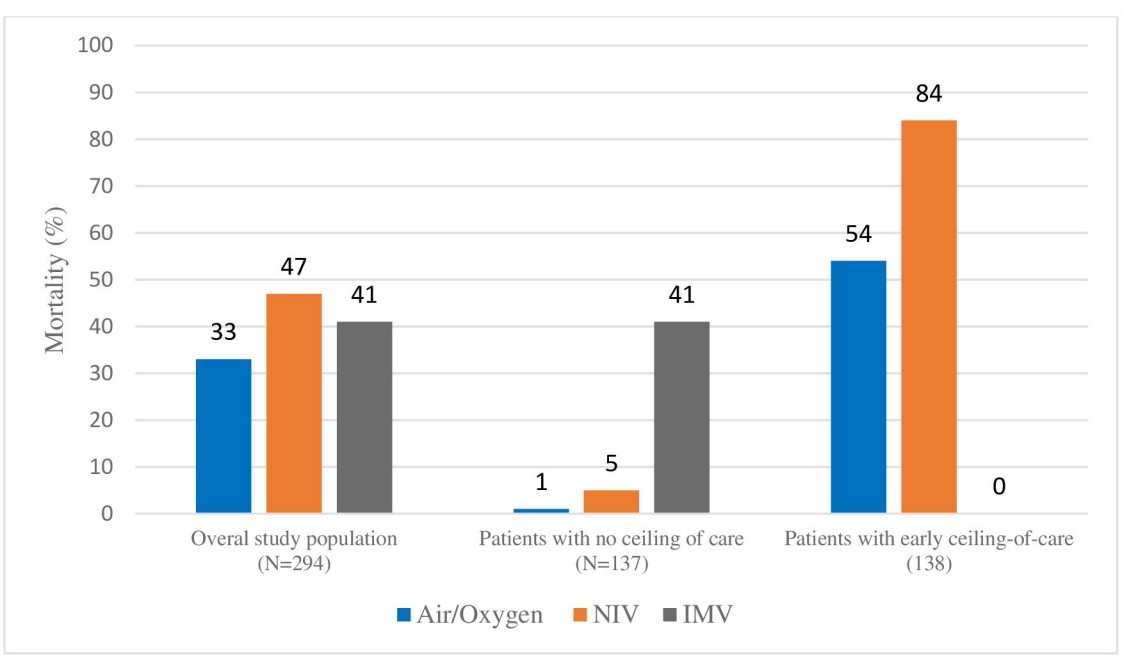

**Fig 1. Mortality of patients by maximal respiratory support received, overall and stratified by ceiling-of-care.**

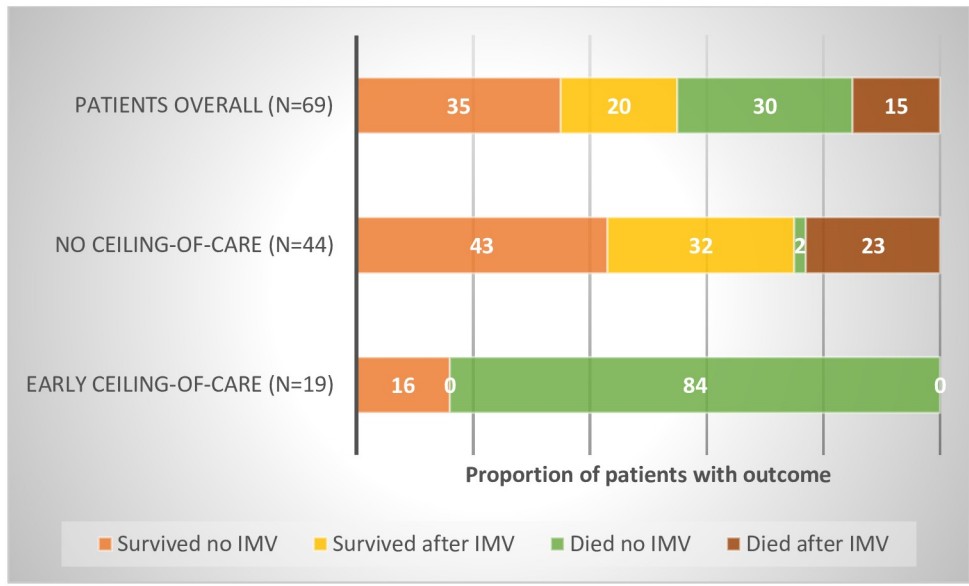

**Fig 2. Outcomes among all patients who ever received CPAP, by ceiling-of-care status (numbered version).**

morbidities, namely hypertension, chronic cardiac disease and dementia. Having an early ceiling-of-care pre-admission/within 24 hours of admission was associated with over 40 times the odds of death in the multivariable analysis (aOR 41.81 95% CI 8.28, 211.17; p<0.001), compared to patients with no ceiling-of-care. Maximal respiratory support with CPAP and IMV was not associated with higher mortality compared to air/oxygen in the crude analysis (OR 1.77 95% CI 0.93, 3.40 and OR 1.42 (0.68, 2.97), respectively; p = 0.19). After adjusting for confounding factors there was strong evidence for higher mortality among hypoxemic patients who required escalation to CPAP or IMV (aOR 5.24 95% CI 1.38, 19.81 and aOR 46.47 95% CI 7.52, 287.08, respectively; p<0.001) compared to air/oxygen, although the confidence intervals were overlapping and were wide especially for IMV, due to the small sample.

## Discussion

In our cohort of 294 hospitalised patients, 69 were trialled on CPAP either as a ceiling-of-care (N = 25), of which only those with an early ceiling-of-care (N = 19) were included in the analysis, or as a potential bridge to IMV (N = 44). Among patients who were trialled on CPAP as a potential bridge to IMV 75% survived—43.2% (N = 19) survived without requiring IMV and a further 31.8% (N = 14) after IMV (Fig 2). Patients who maximally received CPAP also spent significantly less time in hospital compared to IMV patients (median 7 days vs 24 days, $X^2$ p<0.001). The implications of our findings are that a trial of CPAP prior to intubation appears to be an effective treatment strategy in selected patients and may have significant benefits both for patient well-being and health system resources, especially in settings where IMV and accompanying intensive care beds are scarce.

In contrast to patients who had CPAP as a potential bridge to IMV, a high mortality was observed among patients on CPAP as a ceiling-of-care (Fig 1), reflecting both severity of disease, as well as frailty, age and co-morbidities in this cohort [11,12].

Overall, in patients with a ceiling of care, mortality was 5 times higher in those requiring CPAP than in patients treated maximally with air/oxygen after adjusting for confounding factors, importantly age. This higher relative mortality is to be expected, given that the indication

**Table 3. Characteristics of patients who died and predictors of mortality.**

| Patients overall | | Proportion died n/N (%) or median (IQR) | Crude odds ratio (OR)[1] (95% Confidence Interval (CI); Likelihood Ratio Test (LRT) p-value) | | Age category adjusted OR[2] (95% CI; LRT p-value) | | Multivariable OR[3] (95% CI; LRT p-value) | |
|---|---|---|---|---|---|---|---|---|
| **Socio-demographic characteristics** | | | | | | | | |
| **Age (y)** | 20–49 | 2/30 (6.7) | - | 0.001 | | | - | <0.001 |
| | 50–59 | 5/41 (12.2) | | | | | | |
| | 60–69 | 15/54 (27.8) | | | | | | |
| | 70–79 | 32/75 (42.7) | 3.72 (1.95,7.11) | | | | 3.33 (1.11,9.86) | |
| | 80–89 | 35/66 (53.0) | 5.65 (2.90,10.99) | | | | 4.76 (1.44,15.79) | |
| | ≥ 90 | 17/21 (80.1) | 21.25 (6.52,69.26) | | | | 15.72 (3.15,78.47) | |
| **Sex** | Female | 36/111 (32.4) | - | 0.31 | - | 0.04 | - | 0.13 |
| | Male | 70/183 (38.3) | 1.29 (0.79,1.12) | | 1.78 (1.02,3.12) | | 1.70 (0.85,3.41) | |
| **Ethnicity** | White | 94/239 (39.3) | - | 0.02 | - | 0.71 | - | 0.77 |
| | Asian, Black, Other | 12/55 (21.8) | 0.43 (0.22,0.86) | | 0.86 (0.39,1.90) | | 0.85 (0.29,2.53) | |
| **BMI**[4] | | 26 (22.4,30) | 0.95 (0.90,0.99) | 0.03 | 0.98 (0.94,1.04) | 0.56 | 1.00 (0.94,1.07) | 0.87 |
| **Care home resident** | | 37/65 (56.9) | 3.06 (1.74,5.40) | <0.001 | 1.63 (0.86,3.07) | 0.13 | 1.51 (0.69,3.33) | 0.30 |
| **Pre-existing comorbidities** | | | | | | | | |
| **Hypertension** | | 47/109 (43.1) | 1.62 (0.99,2.64) | 0.05 | 1.29 (0.75,2.21) | 0.36 | 1.16 (0.58,2.32) | 0.67 |
| **Chronic cardiac disease** | | 48/90 (53.3) | 2.88 (1.72, 4.81) | <0.001 | 1.43 (0.79,2.59) | 0.23 | 1.18 (0.57,2.43) | 0.66 |
| **Diabetes mellitus without complications** | | 23/56 (41.1) | 1.30 (0.72, 2.36) | 0.39 | 1.30 (0.67,2.49) | 0.44 | 1.15 (0.51,2.66) | 0.73 |
| **Diabetes mellitus with complications** | | 13/26 (50.0) | 1.89 (0.84, 4.25) | 0.12 | 3.10 (1.27,7.55) | 0.01 | 2.73 (0.87,8.52) | 0.09 |
| **Dementia** | | 32/56 (57.1) | 2.95 (1.63,5.36) | <0.001 | 1.45 (0.74,2.85) | 0.28 | 1.48 (0.68,3.24) | 0.32 |
| **Chronic pulmonary disease** | | 21/49 (42.9) | 1.41 (0.76,2.63) | 0.28 | 1.13 (0.57,2.23) | 0.73 | 0.91 (0.41,2.04) | 0.82 |
| **Chronic neurological disorder** | | 17/38 (44.78) | 1.52 (0.76,3.03) | 0.24 | 1.39 (0.65,2.95) | 0.39 | 1.15 (0.45,2.96) | 0.76 |
| **Asthma** | | 9/34 (26.5) | 0.61 (0.27,1.35) | 0.22 | 0.77 (0.33,1.82) | 0.56 | 0.85 (0.30,2.42) | 0.76 |
| **Chronic kidney disease** | | 13/28 (46.4) | 1.61 (0.74,3.53) | 0.23 | 0.94 (0.40,2.23) | 0.89 | 1.20 (0.43,3.33) | 0.73 |
| **Rheumatological condition** | | 6/22 (27.3) | 0.65 (0.24,1.70) | 0.38 | 0.46 (0.16,1.31) | 0.15 | 0.57 (0.17,1.94) | 0.37 |
| **Malignant neoplasm** | | 7/20 (35.0) | 0.96 (0.37,2.49) | 0.94 | 0.64 (0.23,1.76) | 0.39 | 0.64 (0.18,2.19) | 0.47 |
| **Clinical features at admission** | | | | | | | | |
| **Temperature (degrees Celsius)** | | 36.7 (36.3,37.5) | 0.82 (0.65,1.03) | 0.09 | 1.00 (0.77,1.29) | 0.99 | 0.84 (0.58,1.23) | 0.38 |
| **Respiratory rate (breaths per minute)** | | 24 (21,30) | 1.08 (1.04,1.12) | <0.001 | 1.12 (1.07,1.18) | <0.001 | 1.09 (1.03,1.15) | 0.003 |
| **Heart rate (beats per minute)** | | 92 (81,107) | 1.01 (0.99,1.02) | 0.38 | 1.02 (1.00,1.03) | 0.02 | 1.01 (0.99,1.02) | 0.47 |
| **Management** | | | | | | | | |
| **Ceiling-of-care pre-admission or within 24h**[5] | | 80/138 (58.0) | 10.43 (5.60,19.42) | <0.001 | 6.91 (3.09,15.46) | <0.001 | 41.81 (8.28,211.17) | <0.001 |
| **Maximal respiratory support** | | | | | | | | |
| **Air/oxygen** | | 68/215 (31.6) | - | 0.19 | - | <0.001 | | <0.001 |
| **CPAP** | | 21/45 (46.7) | 1.77 (0.93,3.40) | | 5.41 (2.32,12.64) | | 5.24 (1.38,19.81) | |
| **IMV** | | 14/34 (41.2) | 1.42 (0.68, 2.97) | | 7.86 (2.93,21.08) | | 46.47 (7.52, 287.08) | |

[1] Logistic regression models were generated whereby for continuous variables the ORs are for each unit increase in value of the given characteristic; for binary variables the reference category was the absence of a given characteristic; for other categorical variables the reference category is indicated in the table; where there were low number of deaths, categories are combined as shown for age and ethnicity;

[2] Adjusted for age category as in 4 categories: 20-69y, 70-79y, 80-89y, ≥90y;

[3] Adjusted for age category, sex, ceiling-of-care pre-admission or within 24h, diabetes mellitus with complications (ie end organ damage) and maximal respiratory support received;

[4] Missing values for BMI (N = 82) were excluded;

[5] Patients who had a ceiling-of-care introduced after failure to respond to treatment were excluded (N = 19).

for CPAP in these patients is hypoxemia not controlled on standard oxygen therapy, due to more severe Covid-19. Mortality among patients with a ceiling-of-care was over 40 times higher than among those without any limits to their potential treatment pathways even after adjusting for age and other confounding factors, likely reflecting treatment limitations, pre-existing frailty and the burden of other co-morbidities in these patients, which we were not able to adjust for.

Existing evidence for CPAP in severe acute respiratory distress is conflicting, and much of it was initially based on non-Covid-19 pathology [13]. A study examining CPAP use in SARS found that CPAP avoided IMV in 70% of patients [14]. Conversely, a study of non-invasive ventilation in critically ill patients with MERS found that over 90% of patients initially treated with non-invasive ventilation required intubation [15]. Studies from Wuhan, China demon-strated a mortality after CPAP of 44–72% with Covid-19 [16,17]. The extent to which the high mortality related to pre-morbid frailty, treatment limitations, severity of disease or a combina-tion is unclear. UK guidelines recommend that when ICU capacity is limited, CPAP (as the preferred form of non-invasive ventilatory support) should be trialed in COVID-19 patients who remain hypoxemic despite standard oxygen therapy [18]. Data from the ISARIC/WHO CCP-UK study on >20,000 Covid-19 patients from 208 acute hospitals across the country indicate that 16% of patients were trialed on CPAP during admission for Covid-19, which is similar to our study [2]. In contrast, data from the US suggest that as little as 1–2% of patients admitted to ICU had a trial of CPAP prior to intubation [3,19].

Alternative modes of non-invasive respiratory support may be superior to CPAP delivered by face mask [20]. In our study, only 14 patients had CPAP with a helmet. A previous rando-mised trial has demonstrated superiority of CPAP with helmets over CPAP via face masks in avoiding endotracheal intubation (62% vs. 18%) in non-Covid pathology [21]. High-flow oxy-gen through nasal cannula in acute hypoxemic respiratory failure has also been shown to be effective, but the high oxygen consumption associated with this method may present a serious challenge with any surge in demand, hence this was not widely used in the UK [22]. The increased infectious risk posed to healthcare workers from administering non-invasive ventila-tion modalities such as CPAP also requires further research [14,23]. Randomized controlled trials are underway to identify the most effective form of non-invasive pressure support in Covid-19 in reducing the need for intubation [24,25]. Studies are also emerging globally con-firming the usefulness of ward-based CPAP for Covid patients [26,27]. In the meantime, our study provides important insights into the potential utility and limitations of CPAP.

The sample size in this study was relatively small. However, characteristics of our study population were similar to the ISARIC/WHO CCP-UK study data across key features includ-ing sex, age, prevalence of co-morbidities and mortality, suggesting that our patients are representative of UK Covid-19 patients [2]. We have not accounted for other treatments our patients received, including as part of the RECOVERY trial which most patients were enrolled in [28]. However, as the trial randomised treatment allocation it is unlikely it would have sys-tematically biased our findings. We have been able to provide novel data on ceilings-of-care and its impact on outcomes from Covid-19. We also had final outcomes on almost all our patients with just 4 still hospitalised in contrast to many other clinical cohorts on Covid-19 [3,17,19,29].

Our data indicate care home residents are not more likely to die after accounting for age in our study. This is re-assuring with regard to the clinical treatment received in hospital by this sub-group of patients against the background of the concerning public health data on the high death toll from Covid-19 among care home residents [30]. Almost a fifth of our patients were of Asian ethnicity, but these patients were on average younger than the majority white popula-tion, and it was not associated with increased mortality in our study.

The global scale of the pandemic has the potential to overwhelm many health systems, especially ICU resources. There is an urgent imperative for health systems faced with the prospect of shortage of ventilators to harness the role of ventilator-sparing strategies in the treatment of Covid-19 [31–33]. Our data suggest that among patients with no ceiling-of-care, an initial trial of CPAP in selected patients seems a reasonable therapeutic strategy and may potentially delay or even avert the need for intubation in some patients. In contrast, our findings indicate that the use of CPAP among patients with a ceiling-of care has a high failure rate, and it should be used judiciously in close consultation with patients and their representatives where feasible. Further research on this matter would help future management. Our data highlights the importance of accounting for ceilings of respiratory support when interpreting mortality from Covid-19.

This study has provided novel evidence on the respiratory support of patients with Covid-19 and could have significant implications for patients and health systems faced with the unprecedented impact of the pandemic.

## Supporting information

**S1 Fig. Conceptual framework of relationship between exposure and confounding variables and mortality.**
(DOCX)

**S2 Fig. Flow diagram of patients included in the study and mortality by mode of respiratory support and ceiling-of-care.**
(DOCX)

## Acknowledgments

We would like to firstly acknowledge the patients and families of patients, for the ordeal they went through and for the many who have been affected by the pandemic. We would also like to thank the institution and staff of Calderdale and Huddersfield NHS Foundation Trust for their continued help and support with this project, in particular the hard work and dedication of colleagues who gave up their valuable free time to acquire data and help with this project: Dr Joshua Storton, Dr Anneka Biswas, Dr Charlotte Spencer, Dr Rehima Aslam, and Dr Mohammad Abdalmohsen.

## Author Contributions

**Conceptualization:** Jonathan Walker, Kalpana Sabapathy.

**Data curation:** Jonathan Walker, Shaman Dolly, Liji Ng, Melissa Prior-Ong, Kalpana Sabapathy.

**Formal analysis:** Jonathan Walker, Shaman Dolly, Liji Ng, Melissa Prior-Ong, Kalpana Sabapathy.

**Funding acquisition:** Kalpana Sabapathy.

**Investigation:** Jonathan Walker, Kalpana Sabapathy.

**Methodology:** Jonathan Walker, Kalpana Sabapathy.

**Project administration:** Jonathan Walker, Kalpana Sabapathy.

**Resources:** Jonathan Walker, Kalpana Sabapathy.

**Software:** Jonathan Walker, Kalpana Sabapathy.

**Supervision:** Jonathan Walker.

**Validation:** Jonathan Walker, Shaman Dolly, Liji Ng, Melissa Prior-Ong, Kalpana Sabapathy.

**Visualization:** Jonathan Walker, Liji Ng, Kalpana Sabapathy.

**Writing – original draft:** Jonathan Walker, Kalpana Sabapathy.

**Writing – review & editing:** Jonathan Walker, Shaman Dolly, Liji Ng, Melissa Prior-Ong, Kalpana Sabapathy.

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
