## [Decision Letter · Decision Letter 0]

2 Sep 2020

PONE-D-20-20968

The role of CPAP as a potential bridge to invasive ventilation and as a ceiling-of-care for patients hospitalised with Covid-19 - an observational study.

PLOS ONE

Dear Dr. Walker,

Thank you for submitting your manuscript to PLOS ONE. After careful consideration, we feel that it has merit but does not fully meet PLOS ONE’s publication criteria as it currently stands. Therefore, we invite you to submit a revised version of the manuscript that addresses the points raised during the review process.

As suggested by rev n.3, the results section needs deep improvement. It is very hard to read and understand due to a mix of results regarding patients with treatment limitations and patients with no limitations.

As suggested by rev n.1, the effectiveness of CPAP need to be better discussed in light of the more clear results. Is it effective? Which phenotype of patients may benefit more? is it really futile (too expensive/untolerable/not effective) in patients with treatment limitations?

We look forward to receiving your revised manuscript.

Kind regards,

Andrea Coppadoro

Academic Editor

PLOS ONE

Journal Requirements:

2. Please confirm in your methods section and ethics statement that the 'Calderdale and Huddersfield NHS Foundation Trust Research and Development Department' consists of a committee of experts that reviewed and approved your study.

3. In your ethics statement in the manuscript and in the online submission form, please provide additional information about the patient records used in your retrospective study. Specifically, please ensure that you have discussed whether all data were fully anonymized before you accessed them.

4. Please note that PLOS does not permit references to “data not shown.” Authors should provide the relevant data within the manuscript, the Supporting Information files, or in a public repository. If the data are not a core part of the research study being presented, we ask that authors remove any references to these data.

5. To comply with PLOS ONE submission guidelines, in your Methods section, please provide additional information regarding your statistical analyses. For more information on PLOS ONE's expectations for statistical reporting, please see https://journals.plos.org/plosone/s/submission-guidelines.#loc-statistical-reporting.

Reviewers' comments:

Reviewer's Responses to Questions

**Comments to the Author**

1. Is the manuscript technically sound, and do the data support the conclusions?

Reviewer #1: Partly

Reviewer #2: Partly

Reviewer #3: Yes

2. Has the statistical analysis been performed appropriately and rigorously? 

Reviewer #1: Yes

Reviewer #2: Yes

Reviewer #3: Yes

3. Have the authors made all data underlying the findings in their manuscript fully available?

Reviewer #1: No

Reviewer #2: Yes

Reviewer #3: Yes

4. Is the manuscript presented in an intelligible fashion and written in standard English?

Reviewer #1: Yes

Reviewer #2: Yes

Reviewer #3: Yes

5. Review Comments to the Author

Reviewer #1: In this manuscript, the authors publish their experience with COVID patients, focusing specifically on 2 variables: amount of respiratory support (oxygen alone, CPAP, or invasive mechanical ventilation, or IMV), and outcomes for patients with an existing ceiling of care. While this was a descriptive, observational study, the authors sought to describe both of these presumably because of concern about resource use in the context of COVID, as well as the debate that surrounded early versus late intubation for hypoxemic patients.

The setting is 2 large, linked hospitals in England, under the NHS system. The authors collected a significant amount of demographic and vital sign data and attempted to identify factors associated with outcomes as a function of either ceiling-of-care orders or respiratory level of support.

Overall, while I appreciate that these two issues are interrelated, they are really quite distinct and merging them into a single manuscript complicates interpretation of either problem individually. However, this work clearly demonstrates that the use of CPAP as a bridge to intubation (or as a destination) remains a very reasonable practice, with a very acceptable mortality rate, particularly among those without a prespecified ceiling of care; this is important because it overturns the notion that we should proceed to early intubation of COVID patients (which has largely been abandoned at this time) rather than attempting therapy with NIV or high flow oxygen. I am less confident about their interpretations of ceilings-of-care data, as I will outline below. In addition, the authors overinterpreted or overstated their data in several instances.

Major Issues:

• They argue that, because patients with a ceiling-of-care have an 85% mortality when CPAP is used that the “use of CPAP may be futile.” The determination of futility should account for the amount of resources used and the discomfort to the patient; if CPAP can be delivered without a significant resource drain in a relatively comfortable manner, the potential benefit (that it may save lives) may be worthwhile.

• The authors do not seem to account for the fact that patients with an existing ceiling-of-care likely have multiple medical comorbidities that increase their risk of dying of any insult, COVID or otherwise; thus, the description of mortality among these patients is inherently biased towards higher mortality for those with a ceiling of care, unless some attempt is made to control for the comorbidities that they have. This could be accomplished using accepted comorbidity scales like Elixhauser or Charlson. By extension, it’s hardly surprising that patients with an early ceiling-of-care have higher mortality in multivariable analysis. Similarly, the fact that patients who were trialled on CPAP as a potential bridge to IMV and had 43% survival without requiring IMV likely reflects the fact that those without ceiling-of-care limitations were younger and healthier.

• The authors never provide any data attempting to quantify severity of illness on admission, using accepted scoring systems like SOFA, APACHE, or MEWS. Among other things, controlling for severity of illness in their multivariable model would allow them to quantitatively account for the fact that patients being admitted with ceiling-of-care limitations may well have been sicker, owing to their comorbid illnesses that prompt initiation of such limitations in the first place

• They provide no data about the magnitude of therapy provided with CPAP: were patients titrated up to a certain level? What was the median amount of CPAP support required (which would be directly related to severity of hypoxema)? Did they use Bipap at all? The amount of support provided by 5 of CPAP versus 18/12 of Bipap is quite significant. Was there any variation in determination of “failure” of CPAP and subsequent intubation?

Minor Issues:

• The authors note that the “aim of our study was to evaluate the outcomes of patients… treated according to national guidelines.” Was the standard of care constant over the entire time period? As the authors note, initially, there were recommendations to avoid CPAP and proceed to intubation, and while the NHS recommendations changed in early April, patients were admitted from early March to late April, so the standard of care could conceivably have changed over that time.

• They did not prove that use of CPAP allowed them to avert intubations, and the authors should avoid suggesting this; this analysis could be attempted using a retrospective case-control study, where controls and cases are otherwise matched for severity of illness, demographics, etc and the sole difference is whether or not they were intubated.

Reviewer #2: Overall Impression

This study describes the demographics and outcomes of patients admitted to two acute hospitals and describes characteristics of patients by maximal respiratory support received and characteristics of patients who died.

Outcomes amongst patients that were for full escalation and those that had a ceiling of care (CPAP or oxygen) in place within the first 24 hours of their admission are described.

The numbers of patients that received CPAP treatment are small (19 as ceiling of care, 44 as a potential bridge to IMV.

The majority of the manuscript describes patient characteristics and predictors of mortality. The difference in mortality amongst those who received CPAP as a ceiling of care and those that received CPAP but were deemed eligible for IMV if the treatment did not work are stark (25% v 84%) but the numbers are small.

I think the message is an important one (and fits with our experience). Differences (if present) between the location of care need to be highlighted, as do numbers of patients in each group – For IMV/Full escalation (Number of patients that had air/02, CPAP pre IPPV, IPPV without CPAP, and CPAP) and patients with a ceiling of care (numbers that had air/oxygen and CPAP).

Major points

1) A flow diagram would better illustrate numbers in different groups. The key comparison should be between a) those that were for full escalation that received CPAP and b) those that had a ceiling of care in place that received CPAP. The mortality % in these groups is shown in figure 2 but the numbers (although small) should be made clearer. A flow diagram would be useful.

2) The authors mention that CPAP was largely delivered on general wards. This requires further clarification. Were a greater proportion of those patients that were for IMV treated with CPAP on ITU? If so the location of the treatment/expertise of the staff could be important factors in the differences in mortality.

3) Some clarification regarding the numbers in the different groups would be welcome (eg Line 272-274, 19+44 = 63 rather than 69). This might be facilitated by addressing 1).

Minor points

1) Line 349 – The trials explore the role of NIV/CPAP in reducing the need for (rather than prior to) intubation.

2) Typos in hypertext links (. Missing after www) in several references – eg 12, 28.

Reviewer #3: Overall this is a helpful study that adds to the growing body of evidence of CPAP use in CVOID-19. It is quite wordy for the amount of data it presents but some of that is style preference.

1)Abstract:

Minor: You switch between the use of numbers i.e: 347 in line 48 and words for numbers i.e: Two hundred in line 49.

I would suggest sticking to one format, ideally numbers for everything over 10 and words under 10. This is an issue

throughout the manuscript.

Minor: How many people actually started on CPAP would be helpful to know in abstract. It is nicely worded in line 304 of the discussion. It is slightly confusing for the reader when you state 45 on CPAP and 34 on IMV (but actually some of these 34 started on CPAP they did not straight to IMV)

Minor: Line 54: This line seems to imply that CPAP or IMV were somehow the reason for higher mortality? Is this correct or in fact they are sicker patients compared to those on oxygen. Can I suggest if the later is the case this is reworded, it could imply it is better not to escalate care which I am not sure is your meaning.

2) Introduction.

This is well written. No major problems.

Minor: I am interested you have not mention high flow oxygen anywhere. Why is this? It maybe worth one line in the introduction or methods saying high flow oxygen was not available or not used due to oxygen supply or just not considered etc.

3) Methods

Appears clear and make sense to the reader.

Major: One of the reason that CPAP was considered a risk at the start of COVID was it considered an AGP. There is no mention in your methods other than "largely on wards" of where this took place, what was the nursing ratio?, were extra safety measures used? , did you use negative pressure rooms?, was it only those with ceiling of care that had CPAP on wards, could the place they had CPAP have made a difference to the outcome, Did staff wear full AGP PPE? This document is already wordy heavy you could reduce you introduction or results sections and add a few lines about this in the methods. You may also wish to add a line in the results describing if there were adverse events from the CPAP or suddenly an increase in sickness level among staff that you know about.

Major: There seems to be limited detail on the CPAP give, what pressures where used? Did you have protocol? How much oxygen was used etc.

Minor: Do you have reference number from R&D re: Ethic approval or was it registered as service evaluation?

4) Results: This is the most unclear section of manuscript and need some major revision to make it flow.

If you are going to put key findings at the start of results: Keep them as key finding, Line 183 under key findings describes their characteristics yet this is under key findings. I don't think age and sex of the patients etc is key finding.

Personally I would suggest starting with the numbers in the study then move on to the participant characteristics, key findings can be highlighted in the first section of the discussion. Then have you sub-section of results. I would suggest there is clear section of CPAP as a bridge to IMV (or preventing IMV) and a section on CPAP in those with ceiling care and factors effecting mortality. These are just suggestions but I would like to see the section re worked.

Although it needs stating I would have thought you would expect those with a ceiling care who are on CPAP to have a higher mortality rate as they are logically sicker? The way this results section is written could imply the CPAP contributed to the outcome- if it did this need discussion in the next section, if fact these patient were just sicker this needs some rewording- in discussion you suggest it is due to severity.

It would be helpful to know how much oxygen & what CPAP these patients were on? And how long did it take patients to fail on CPAP and need IMV.

The clinical picture is also not 100% clear, although there are lots of pre-existing condition accounted for, do you have ABG's, is there an P/F ratio you could give us?

5) The discussion is clearer than results.

Major: Line 300. You are not not the first paper to describe CPAP as bridge to IMV. There are several paper from Italy and some from the UK describing similar. You maybe the first describing ceiling care although there are papers from Italy with very similar cohort. Some examples (and there are more): https://erj.ersjournals.com/content/early/2020/07/30/13993003.02130-2020

https://papers.ssrn.com/sol3/papers.cfm?abstract_id=3566170

https://bmjopenrespres.bmj.com/content/7/1/e000621.abstract
https://bmjopenrespres.bmj.com/content/7/1/e000639.abstract

https://www.medrxiv.org/content/10.1101/2020.06.05.20123307v1

https://www.ncbi.nlm.nih.gov/pmc/articles/PMC7190517/

Minor: line 319: Again a similar theme, I think you mean CPAP mortality is higher but it is likely to be because of the severity of disease, this need to be clearer throughout the paper if this is how you interpret the results, this is why some markers of severity would have been nice in the results.

Major: line 328 onward, there is existing evidence (albeit limited) in covid-19. I would suggest you try and compare your results to those of Covid-19 studies as well as MERS and SARS papers. There are a few papers/short reports coming from UK hospital coming from UK hospital now, some are listed above, there may well be more.

Overall once the key message is made clearer and ideally a little more data regarding the severity of patients is added this is a helpful paper that adds to the body of evidence regarding oxygen/CPAP and IMV in COVID. I would suggested this is more of a very helpful case series than novel new research and it maybe better presenting it than way.

6. PLOS authors have the option to publish the peer review history of their article (what does this mean?). If published, this will include your full peer review and any attached files.

Reviewer #1: **Yes: **Philip Verhoef

Reviewer #2: **Yes: **Hassan Burhan

Reviewer #3: No

---

## [Author Response · Author response to Decision Letter 0]

5 Nov 2020

2. Please confirm in your methods section and ethics statement that the 'Calderdale and Huddersfield NHS Foundation Trust Research and Development Department' consists of a committee of experts that reviewed and approved your study.

Response: 

Thank you for raising this. On application for approval, our local R&D Department noted that this project is examining recognised treatment protocols and procedures using internal data, which would be de-identified for the purposes of data analysis outside the institution. As such they considered this project a service evaluation to establish a standard, and as such did not require further approval A clarification about this has been added to the Methods section, in lines 105- 107 where it states “Data analysed was departmental, routinely collected and fully anonymized, and as such The Calderdale and Huddersfield NHS Foundation Trust Research and Development office considered this project a service evaluation to establish a standard, and did not require further approval. “

The study is registered with the Calderdale and Huddersfield NHS Foundation Trust Clinical Governance Department. 

We have also stated in line 126-127 of Methods that “This data was fully anonymised and de-identified prior to inclusion in the analysis and the study.”

3. In your ethics statement in the manuscript and in the online submission form, please provide additional information about the patient records used in your retrospective study. Specifically, please ensure that you have discussed whether all data were fully anonymized before you accessed them. 

Response: 

Thank you; we have highlighted this in the text. Please see response to point 2 above. We can confirm that all data was fully anonymised and de-identified prior to inclusion in the analysis and the study. We have changed the wording to read “This data was fully anonymised and de-identified prior to being accessed for inclusion in the analysis and the study” in lines 126- 127. This is now highlighted in the ethics statement of the online submission form. The Trust Caldicott guardian has approved the use of data for this purpose. 

4. Please note that PLOS does not permit references to “data not shown.” Authors should provide the relevant data within the manuscript, the Supporting Information files, or in a public repository. If the data are not a core part of the research study being presented, we ask that authors remove any references to these data.

 Response: 

Thank you for the guidance. We have now removed any reference to data which is not reproducible from our dataset.

5. To comply with PLOS ONE submission guidelines, in your Methods section, please provide additional information regarding your statistical analyses. For more information on PLOS ONE's expectations for statistical reporting, please see https://journals.plos.org/plosone/s/submission-guidelines.#loc-statistical-reporting.

Response: 

Thank you for the helpful link. We have ensured our final draft complies with the above guidelines.

Reviewer #1: In this manuscript, the authors publish their experience with COVID patients, focusing specifically on 2 variables: amount of respiratory support (oxygen alone, CPAP, or invasive mechanical ventilation, or IMV), and outcomes for patients with an existing ceiling of care. While this was a descriptive, observational study, the authors sought to describe both of these presumably because of concern about resource use in the context of COVID, as well as the debate that surrounded early versus late intubation for hypoxemic patients.

The setting is 2 large, linked hospitals in England, under the NHS system. The authors collected a significant amount of demographic and vital sign data and attempted to identify factors associated with outcomes as a function of either ceiling-of-care orders or respiratory level of support.

Overall, while I appreciate that these two issues are interrelated, they are really quite distinct and merging them into a single manuscript complicates interpretation of either problem individually. However, this work clearly demonstrates that the use of CPAP as a bridge to intubation (or as a destination) remains a very reasonable practice, with a very acceptable mortality rate, particularly among those without a prespecified ceiling of care; this is important because it overturns the notion that we should proceed to early intubation of COVID patients (which has largely been abandoned at this time) rather than attempting therapy with NIV or high flow oxygen. I am less confident about their interpretations of ceilings-of-care data, as I will outline below. In addition, the authors overinterpreted or overstated their data in several instances.

Major Issues:

• They argue that, because patients with a ceiling-of-care have an 85% mortality when CPAP is used that the “use of CPAP may be futile.” The determination of futility should account for the amount of resources used and the discomfort to the patient; if CPAP can be delivered without a significant resource drain in a relatively comfortable manner, the potential benefit (that it may save lives) may be worthwhile.

Response:

We thank the reviewers for raising this point, and we have clarified the wording in line 64 of the Abstract to read “CPAP seems to offer little additional survival benefit beyond oxygen therapy alone” in this group. Intolerance to CPAP is well described in the literature and is also re-enforced in local UK guidelines where it states that “hoods and masks can be distressing for patients” and that “CPAP should not be used in those with agitation and confusion.” (Ref 9 in the manuscript). 

The reviewer is correct that a trial of CPAP may be worthwhile if it can be delivered in a relatively comfortable manner, but survival rate in those with a DNACPR is comparable to those treated without CPAP, and especially in the setting of confusion and agitation CPAP can result in significant distress. For this reason, its use in this population group should be carefully considered. 

• The authors do not seem to account for the fact that patients with an existing ceiling-of-care likely have multiple medical comorbidities that increase their risk of dying of any insult, COVID or otherwise; thus, the description of mortality among these patients is inherently biased towards higher mortality for those with a ceiling of care, unless some attempt is made to control for the comorbidities that they have. This could be accomplished using accepted comorbidity scales like Elixhauser or Charlson. By extension, it’s hardly surprising that patients with an early ceiling-of-care have higher mortality in multivariable analysis. Similarly, the fact that patients who were trialled on CPAP as a potential bridge to IMV and had 43% survival without requiring IMV likely reflects the fact that those without ceiling-of-care limitations were younger and healthier.

Response: 

This is correct, and we have highlighted this very point in line 357- 352 of the Discussion where we state “a high mortality was observed among patients on CPAP as a ceiling-of-care (Figure-1), reflecting both severity of disease, as well as frailty, age and co-morbidities in this cohort.”

Similarly, in line 367- 374 we state that a “higher relative mortality is to be expected given that the indication for escalation of respiratory support to CPAP was due to greater severity of Covid-19. Mortality among patients with a ceiling-of-care was over 40 times higher than among those without any limits to their potential treatment pathways even after adjusting for age and other confounding factors, reflecting the limited treatment received, residual effects of pre-existing frailty and burden of other pre-morbidities in patients with an early ceiling-of-care which we were not able to adjust for.”

 The relationship between co-morbidities and outcomes in Covid has been described in many other studies, such as Docherty (Ref 2) as well as the ICNARC data (Ref 5), and although co-morbidities individually and collectively are associated with increased mortality, it is clear from these and other studies that, at the time of this study, age was by far the single most influential determinant of outcome, and this finding was also reflected in our data. Since then, however, other studies have identified frailty as an even better prognostic indicator than age:

National UK Guidelines (NICE) specified from the start of the pandemic that “on admission to hospital” doctors should “assess all adults for frailty” (Ref 10)

This approach is limited but pragmatic and was adopted at our Trust from the beginning. The outcome of adhering to UK guidelines is that all patients admitted with Covid were naturally separated into two distinct cohorts of patients: those deemed by the treating doctors to be too frail for invasive ventilation, and those who were deemed less frail. 

 Ceiling of Care decision were thus made in accordance with UK guidelines mainly on the basis of frailty, rather than comorbidities. NICE Guidelines state that medics should “Sensitively discuss a possible 'do not attempt cardiopulmonary resuscitation' decision with all adults with capacity and an assessment suggestive of increased frailty (for example, a CFS score of 5 or more).”

 This study examined the quantitative differences in outcome between these two different cohorts, independent of comorbidity scales such as Elixhauser or Charlsson, which currently and at the time of the pandemic played no formal role in escalation decisions within the UK. 

The higher mortality seen in patients who were assessed as being frail was certainly expected, but at the time of this study their outcomes and response to treatment relative to non-frail patients had not been quantified. 

• The authors never provide any data attempting to quantify severity of illness on admission, using accepted scoring systems like SOFA, APACHE, or MEWS. Among other things, controlling for severity of illness in their multivariable model would allow them to quantitatively account for the fact that patients being admitted with ceiling-of-care limitations may well have been sicker, owing to their comorbid illnesses that prompt initiation of such limitations in the first place

Response: 

Thank you. The reviewers are correct in pointing out that quantifying severity of illness would enrich the data. However, the SOFA and APACHE scoring systems require arterial blood gasses which were not indicated in the majority of our cohort. Indeed, in the UK, national guidelines specify that “ unless there are reasons to suspect CO2 retention, arterial lines/blood gases are not needed, and patients can be monitored using continuous peripheral arterial oxygen saturation (SpO2) with an appropriate level of nursing support” (Ref 9 in the manuscript).

On our EPR, oxygen saturations prior to oxygen administration were not reliably captured in the majority of cases. 

Oxygen saturations were usually recorded once the target saturations had been achieved, meaning this data was homogenous (with nearly all oxygen saturations documented as being over 92%), with oxygen requirements as the only variable parameter. Thus we had to rely on oxygen requirements as a proxy measure for hypoxia. Other observations, such as heart rate and blood pressure, were used in the study and these have been documented in Table 3. 

• They provide no data about the magnitude of therapy provided with CPAP: were patients titrated up to a certain level? What was the median amount of CPAP support required (which would be directly related to severity of hypoxema)? Did they use Bipap at all? The amount of support provided by 5 of CPAP versus 18/12 of Bipap is quite significant. Was there any variation in determination of “failure” of CPAP and subsequent intubation?

Response:

Thank you. We have now clarified this in the Procedures section of Methods, line 135-137. We have written:

“Clinical management was according to the NHS Specialty specific guidelines (Ref 9). These state that “CPAP is the primary mode of non-invasive respiratory support for hypoxic COVID19 patients. Suggested initial settings are 10 cmH2O + 60% oxygen.” 

In this study, patients not maintaining oxygen saturations over either 92% or 94% on 40 - 60% oxygen via a venturi were commenced on CPAP 10 cm H20 and 10 litres oxygen, according to physician discretion. 

This could be adjusted as required, to a maximum of 15 cm H2O and 15 litres O2.”

The text clearly states that the patients included in the study were on CPAP only, and not BiPAP. As an interesting aside – and not included in this study - BiPAP was used on only three patients: in two patients BiPAP was administered on the ward and titrated to oxygen saturations of 88-92%, as a ceiling of care. One survived. Another had BiPAP as a bridge on ITU, and was subsequently intubated, but did not survive. 

Minor Issues:

• The authors note that the “aim of our study was to evaluate the outcomes of patients… treated according to national guidelines.” Was the standard of care constant over the entire time period? As the authors note, initially, there were recommendations to avoid CPAP and proceed to intubation, and while the NHS recommendations changed in early April, patients were admitted from early March to late April, so the standard of care could conceivably have changed over that time.

Thank you, that is a good point. Internationally, guidelines certainly changed considerably over this time, but the standard of care pertinent to CPAP use within our hospitals was constant over this time period. UK Specialty guides on the use of non-invasive respiratory support in adult patients with coronavirus Version 2 was published on 26th March 2020. This outlined the treatment strategy: If RR ≥ 20bpm with SpO2≤94% on FiO2 >40%, start NRB oxygen and senior review to consider starting a “trial CPAP 10cm H2O with FiO2 0.6” 

Our local guidelines had used the same protocol prior to Version 2, with the exception that our CPAP devices used entrained oxygen of 10 litres on initiation, rather than an FiO2 of 0.6, and in practice CPAP was often not started until O2 saturations were 92% or less on 40% O2 via a venturi. 

Version 3 published on the 3rd April did not alter this. 

With regards to clinical decisions on ceilings of care, the National Institute for Clinical Excellence (NICE) sets many of the clinical guidelines within the UK. 

NICE COVID-19 rapid guideline: critical care in adults was published on 20th March 2020. This recommended that healthcare professionals “sensitively discuss a possible 'do not attempt cardiopulmonary resuscitation' decision with all adults with capacity and an assessment suggestive of increased frailty (for example, a CFS score of 5 or more).”

Again, assessment of frailty had been used in our Trust, and many other Trusts, prior to formal nationalisation of the guidelines by NICE, and the standard of care did not change substantially over the course of this study.

• They did not prove that use of CPAP allowed them to avert intubations, and the authors should avoid suggesting this; this analysis could be attempted using a retrospective case-control study, where controls and cases are otherwise matched for severity of illness, demographics, etc and the sole difference is whether or not they were intubate

Response: 

Thank you, we have changed the wording to reflect this point in line 62 – 64 of the introduction, where it now states: “Our data suggest that among patients with no ceiling-of-care, an initial trial of CPAP as a potential bridge to IMV offers a favourable therapeutic alternative to early intubation. In contrast, among patients with a ceiling-of care, CPAP seems to offer little additional survival benefit beyond oxygen therapy alone.”

 It is correct that we did not prove that use of CPAP allowed us to avert intubations. However, at the time this study was undertaken, high flow nasal oxygen was not routinely used in our hospitals or in the UK generally, and so patients who were not able to maintain their oxygen saturations on standard concentrated oxygen therapy alone, essentially had two escalation options: CPAP or intubation, and it is likely that the majority of patients for full escalation on CPAP would have required intubation had CPAP not been available. Within our data, 43% of patients who were trialled on CPAP survived without intubation 

Reviewer #2: Overall Impression

This study describes the demographics and outcomes of patients admitted to two acute hospitals and describes characteristics of patients by maximal respiratory support received and characteristics of patients who died.

Outcomes amongst patients that were for full escalation and those that had a ceiling of care (CPAP or oxygen) in place within the first 24 hours of their admission are described.

The numbers of patients that received CPAP treatment are small (19 as ceiling of care, 44 as a potential bridge to IMV.

The majority of the manuscript describes patient characteristics and predictors of mortality. The difference in mortality amongst those who received CPAP as a ceiling of care and those that received CPAP but were deemed eligible for IMV if the treatment did not work are stark (25% v 84%) but the numbers are small.

I think the message is an important one (and fits with our experience). Differences (if present) between the location of care need to be highlighted, as do numbers of patients in each group – For IMV/Full escalation (Number of patients that had air/02, CPAP pre IPPV, IPPV without CPAP, and CPAP) and patients with a ceiling of care (numbers that had air/oxygen and CPAP).

Major points

1) A flow diagram would better illustrate numbers in different groups. The key comparison should be between a) those that were for full escalation that received CPAP and b) those that had a ceiling of care in place that received CPAP. The mortality % in these groups is shown in figure 2 but the numbers (although small) should be made clearer. A flow diagram would be useful.

Response: 

We agree with the reviewer that a flow diagram is helpful and in fact had already included exactly this figure in the Supplementary material (Supplementary Figure 2). If the editorial team think it is feasible, we would be happy to include this in the main manuscript. 

2) The authors mention that CPAP was largely delivered on general wards. This requires further clarification. Were a greater proportion of those patients that were for IMV treated with CPAP on ITU? If so the location of the treatment/expertise of the staff could be important factors in the differences in mortality.

Response: 

Thank you, that is an important point. Patients who were not for intubation in the event of deterioration were almost exclusively treated on the wards. Only one was treated on ITU. 

About three quarters of Patients treated with CPAP as a bridge to IMV were treated on the wards. Interestingly, mortality was higher if CPAP was delivered on ITU and we think this is because more unstable patients were selected to be treated on ITU. Hence, we thought data relating to location of CPAP delivery is likely to only demonstrate bias and was not included. We do have this data available, and would be happy to include it if the editors advise so, but unfortunately it is not in our original raw data set. 

We have, however, adjusted the wording in lines 141- 142 of the Methods section to read “With only one exception, CPAP as a ceiling of care was started on the respiratory wards in all cases. CPAP as a bridge to IMV was started on the respiratory wards in most cases.”

Factors that might be expected to contribute to an excess mortality - staffing ratios and skill mixes on the wards, for instance - demonstrated wide variability throughout the day and throughout the week. Furthermore, the extent to which a patient could be closely monitored depended on where they were in a ward – for instance, whether they were isolated in a side room or in a highly visible open bay. 

We would not be able to account for these variables in a study of this size, so the location of CPAP was not examined further. 

3) Some clarification regarding the numbers in the different groups would be welcome (eg Line 272-274, 19+44 = 63 rather than 69). This might be facilitated by addressing 1).

Response: 

We thank the reviewer for highlighting this; as is described in Response 1 to Reviewer #2 above, the flow diagram in the Supplementary material (Supplementary Figure 2) does help to clarify the numbers and we would be glad to include it in the main manuscript. The six patients unaccounted for in the above were patients who had a ceiling of care introduced later during the admission. As described in lines 174- 176 of the Statistical analysis “Patients who had a ceiling-of-care instituted more than 24h after admission were omitted as reverse causality was possible, i.e. that a ceiling-of-care was introduced following failure to respond to treatment.” 

. 

Minor points

1) Line 349 – The trials explore the role of NIV/CPAP in reducing the need for (rather than prior to) intubation.

Response:

 Thank you for pointing this out. This has now been corrected, and line 396 - 400 now reads: Randomized controlled trials are underway to identify the most effective form of non-invasive pressure support in Covid-19 in reducing the need for intubation. Studies are emerging globally confirming the usefulness of ward-based CPAP for Covid patients. In the meantime, our study provides important insights into the potential utility and limitations of CPAP.”

2) Typos in hypertext links (. Missing after www) in several references – eg 12, 28.

Response: We thank the reviewers for highlighting these errors and have rectified them.

Reviewer #3: Overall this is a helpful study that adds to the growing body of evidence of CPAP use in CVOID-19. It is quite wordy for the amount of data it presents but some of that is style preference.

1)Abstract:

Minor: You switch between the use of numbers i.e: 347 in line 48 and words for numbers i.e: Two hundred in line 49.

I would suggest sticking to one format, ideally numbers for everything over 10 and words under 10. This is an issue

throughout the manuscript.

Response: 

Thank you. We have adhered to widely accepted convention to use words to cite numbers at the beginning of sentences and numerals thereafter. For numbers up to ten, we also adhere with convention and use words.

Minor: How many people actually started on CPAP would be helpful to know in abstract. It is nicely worded in line 304 of the discussion. It is slightly confusing for the reader when you state 45 on CPAP and 34 on IMV (but actually some of these 34 started on CPAP they did not straight to IMV)

Response: 

Thank you. We have added this to the abstract as suggested. Lines 51 – 53 now state “Two hundred and fifteen patients (73.1%) maximally received air/standard oxygen therapy, and 45 (15.3%) patients maximally received CPAP. Thirty-four patients (11.6%) required IMV, of which 24 had received prior CPAP.”

Minor: Line 54: This line seems to imply that CPAP or IMV were somehow the reason for higher mortality? Is this correct or in fact they are sicker patients compared to those on oxygen. Can I suggest if the later is the case this is reworded, it could imply it is better not to escalate care which I am not sure is your meaning.

Response: 

Thank you. We have adjusted the wording in this section to make this more explicit. Lines 56 – 59 now reads: “Overall, there was strong evidence for higher mortality among patients who required CPAP or IMV, compared to those who required only air/oxygen (aOR 5.24 95%CI: 1.38,19.81 and aOR 46.47 95%CI: 7.52,287.08, respectively; p<0.001), and among patients with early ceiling-of-care compared to those without a ceiling (aOR 41.81 95%CI: 8.28,211.17; p<0.001).”

We have emphasized in the Discussion (current version) and illustrated through Supplementary Figure 1 that higher mortality seen in CPAP and ITU patients, as compared to patients requiring only oxygen, is thought to reflect greater severity of disease necessitating higher levels of therapy. 

2) Introduction.

This is well written. No major problems.

Minor: I am interested you have not mention high flow oxygen anywhere. Why is this? It maybe worth one line in the introduction or methods saying high flow oxygen was not available or not used due to oxygen supply or just not considered etc.

Response

Thank you. We have clarified this in the introduction by stating in lines 86-89 that “The use of CPAP in Covid-19 has been questioned,8 but in contrast to many other healthcare settings, CPAP is used in the UK in preference to High Flow Nasal Oxygen, largely due to concerns about oxygen supplies.”

3) Methods

Appears clear and make sense to the reader.

Major: One of the reason that CPAP was considered a risk at the start of COVID was it considered an AGP. There is no mention in your methods other than "largely on wards" of where this took place, what was the nursing ratio?, were extra safety measures used? , did you use negative pressure rooms?, was it only those with ceiling of care that had CPAP on wards, could the place they had CPAP have made a difference to the outcome, Did staff wear full AGP PPE? This document is already wordy heavy you could reduce you introduction or results sections and add a few lines about this in the methods. You may also wish to add a line in the results describing if there were adverse events from the CPAP or suddenly an increase in sickness level among staff that you know about.

Response: 

Thank you. Like most UK hospitals we had very limited negative pressure rooms (as a matter of fact we had 2) and these were quickly overwhelmed with the Covid pandemic. The location of CPAP, and why this was not explored in greater detail, has been previously explained in response 2 to reviewer #2 above. A fuller description of the respiratory wards has been added to the Procedures of the Method section. 

For Covid cases requiring AGPs such as CPAP within our hospital sites, the priority was to fill negative pressure rooms initially, then once these were full to decant into open bays of proven Covid cases, then full wards. 

Full PPE was worn in the presence of any AGPs, although the exact definition of “full PPE” did vary throughout the study, according to guidelines and availability of resources. It was considered beyond the scope of this study to investigate adverse effects of CPAP (other than failure and death), or the effect of PPE on healthcare workers in the presence of AGPs. 

Major: There seems to be limited detail on the CPAP give, what pressures where used? Did you have protocol? How much oxygen was used etc.

. 

Response: 

We thank the reviewer for highlight this and it has now been explained more fully in the Procedures section of Methods, and explained above in the response to Reviewer #1. 

NHS Guidelines were followed. These state that “CPAP is the primary mode of non-invasive respiratory support for hypoxaemic COVID19 patients. Suggested initial settings are 10 cmH2O + 60% oxygen.”

In this study, local guidelines were followed which closely reflected NHS Guidelines: patients not maintaining oxygen saturations over either 92% or 94% on 40 - 60% oxygen via a venturi were commenced on CPAP 10 cm H20 and 10 litres oxygen, according to physician discretion. 

This could be adjusted, as required, to a maximum of 15 cm H2O and 15 litres O2. Pressure was adjusted in preference to oxygen. 

In total, BiPAP was used on only three patients: in two patients BiPAP was administered on the ward and titrated to oxygen saturations of 88-92%, as a ceiling of care. One survived. Another had BiPAP as a bridge on ITU, and was subsequently intubated, but did not survive. 

Minor: Do you have reference number from R&D re: Ethic approval or was it registered as service evaluation?

Response:

Thank you for raising this. On application for approval, our local R&D Department noted that this project is examining a recognised treatment using internal data, which would be de-identified for the purposes of data analysis outside the institution. As such they considered this project a service evaluation to establish a standard, and as such did not require further approval. A clarification about this has been added to the Methods section, as mentioned in the response above. 

4) Results: This is the most unclear section of manuscript and need some major revision to make it flow.

If you are going to put key findings at the start of results: Keep them as key finding, Line 183 under key findings describes their characteristics yet this is under key findings. I don't think age and sex of the patients etc is key finding.

Personally I would suggest starting with the numbers in the study then move on to the participant characteristics, key findings can be highlighted in the first section of the discussion. Then have you sub-section of results. I would suggest there is clear section of CPAP as a bridge to IMV (or preventing IMV) and a section on CPAP in those with ceiling care and factors effecting mortality. These are just suggestions but I would like to see the section re worked.

Response: 

We thank the reviewers for the helpful feedback and have re-arranged the Results as indicated in track changes.

Although it needs stating I would have thought you would expect those with a ceiling care who are on CPAP to have a higher mortality rate as they are logically sicker? The way this results section is written could imply the CPAP contributed to the outcome- if it did this need discussion in the next section, if fact these patient were just sicker this needs some rewording- in discussion you suggest it is due to severity.

Response: 

Thank you. This same point was also raised by reviewer #1, and we have hopefully addressed this in our response above. 

It would be helpful to know how much oxygen & what CPAP these patients were on? And how long did it take patients to fail on CPAP and need IMV.

The clinical picture is also not 100% clear, although there are lots of pre-existing condition accounted for, do you have ABG's, is there an P/F ratio you could give us?

Response:

Thank you. 

UK national guidelines relating to CPAP in Covid patients specify that “ Unless there are reasons to suspect CO2 retention, arterial lines/blood gases are not needed, and patients can be monitored using continuous peripheral arterial oxygen saturation (SpO2) with an appropriate level of nursing support” (Ref 9 in the manuscript).

On our EPR, oxygen saturations prior to oxygen administration were not reliably captured in the majority of cases. 

Oxygen saturations were usually recorded once the target saturations had been achieved, meaning there was a great deal of homogeneity in this data, which contributed little to the clinical picture. Oxygen requirements were thus the variable parameter which we had to rely on as a proxy measure for hypoxia. 

Similarly, accurate and reliable documentation of when CPAP was initiated and stopped, when breaks were given, when pressures and oxygen were adjusted, and how much proning was effected, was scarce. Although a very interesting question, it was decided we did not have enough reliable data to pursue this line of enquiry. 

5) The discussion is clearer than results.

Major: Line 300. You are not not the first paper to describe CPAP as bridge to IMV. There are several paper from Italy and some from the UK describing similar. You may be the first describing ceiling care although there are papers from Italy with very similar cohort. Some examples (and there are more): 

https://erj.ersjournals.com/content/early/2020/07/30/13993003.02130-2020

https://papers.ssrn.com/sol3/papers.cfm?abstract_id=3566170

https://bmjopenrespres.bmj.com/content/7/1/e000639.abstract

https://www.medrxiv.org/content/10.1101/2020.06.05.20123307v1

https://www.ncbi.nlm.nih.gov/pmc/articles/PMC7190517/

Response. 

We would like to thank the reviewers for these helpful links. Our manuscript was first submitted to PLOS One on June 27th, and we have changed the wording to more accurately reflect the large body of literature that has since been published. We have also referenced a selection of these articles in lines 396- 400 in the manuscript as an update. Here we have stated “Randomized controlled trials are underway to identify the most effective form of non-invasive pressure support in Covid-19 in reducing the need for intubation.( Referenced to 24, 25) Studies are also emerging globally confirming the usefulness of ward-based CPAP for Covid patients.” (Referenced to 26, 27)

Minor: line 319: Again a similar theme, I think you mean CPAP mortality is higher but it is likely to be because of the severity of disease, this need to be clearer throughout the paper if this is how you interpret the results, this is why some markers of severity would have been nice in the results.

Response:

Thank you. This is a point previously raised, and we hope we have fully addressed it in our first response to reviewer #1. 

Major: line 328 onward, there is existing evidence (albeit limited) in covid-19. I would suggest you try and compare your results to those of Covid-19 studies as well as MERS and SARS papers. There are a few papers/short reports coming from UK hospital coming from UK hospital now, some are listed above, there may well be more.

Response: We thank the reviewers for bringing these articles to our attention. Please see response above. This has now been referred to in the manuscript in line 396- 400 where it states “Randomized controlled trials are underway to identify the most effective form of non-invasive pressure support in Covid-19 in reducing the need for intubation. Studies are also emerging globally confirming the usefulness of ward-based CPAP for Covid patients.”

---

## [Decision Letter · Decision Letter 1]

27 Nov 2020

PONE-D-20-20968R1

The role of CPAP as a potential bridge to invasive ventilation and as a ceiling-of-care for patients hospitalised with Covid-19 - an observational study.

PLOS ONE

Dear Dr. Walker,

Thank you for submitting your manuscript to PLOS ONE. After careful consideration, we feel that it has merit but does not fully meet PLOS ONE’s publication criteria as it currently stands. Therefore, we invite you to submit a revised version of the manuscript that addresses the points raised during the review process.

The reviewers positively commented the new version of the manuscript; however, few more changes are requested for the manuscript to be accepted for publication.

- The delivery of CPAP in your study needs to be clarified under many aspects. 1) Which interface was used? In the discussion, you state that 14 patients used helmet, while (I assume) the large part used a tight face mask. Report the exact numbers in the methods section and, if the sentence is correct, reword in the abstract “Sixty-nine patients were trialled on CPAP, mostly delivered by face mask, …”. 2) Which type of system was used to provide flow? Is it a free-flow Venturi, (I guess that’s not the case, since you state “CPAP is used in the UK in preference to High Flow Nasal Oxygen, largely due to concerns about oxygen supplies” and you refer to 10-15 l/min of oxygen consumption) a Bussignac valve system, a ventilator/turbine system? I guess that you used different systems for helmets and masks. Please add all the requested information in the methods section.

- abstract: there are conflicting information: “Sixty-nine patients were trialled on CPAP either as a ceiling of care (N=25) or as a potential bridge to IMV (N=44).” but later you state “Among all patients trialled on CPAP either as a potential bridge to IMV (N=44) or as a ceiling-of-care (N=19) mortality...”. please report only one of the two sentences to avoid confusion (is it 25+44 or 19+44?) and check consistency with figure 2 of the supplements

- abstract: please report some data about use of CPAP in patients without ceiling of care. Add a sentence like “Among patients without ceiling of care (n=AA), a CPAP trial was started in BB, while CC needed prompt intubation; CPAP failure leading to intubation occurred in DD, while EE patients successfully recovered without intubation maximally receiving CPAP”

If I desumed the numbers correctly, it should be: AA=156; BB=44; CC=10; DD about 26; EE=18

- Figure 1: add the raw number of patients for each column

- Figure 2 supplements: add the number of excluded patients (apparently numbers are missing)

We look forward to receiving your revised manuscript.

Kind regards,

Andrea Coppadoro

Academic Editor

PLOS ONE

Reviewers' comments:

Reviewer's Responses to Questions

**Comments to the Author**

1. If the authors have adequately addressed your comments raised in a previous round of review and you feel that this manuscript is now acceptable for publication, you may indicate that here to bypass the “Comments to the Author” section, enter your conflict of interest statement in the “Confidential to Editor” section, and submit your "Accept" recommendation.

Reviewer #1: All comments have been addressed

Reviewer #2: All comments have been addressed

2. Is the manuscript technically sound, and do the data support the conclusions?

Reviewer #1: Yes

Reviewer #2: Yes

3. Has the statistical analysis been performed appropriately and rigorously? 

Reviewer #1: Yes

Reviewer #2: Yes

4. Have the authors made all data underlying the findings in their manuscript fully available?

Reviewer #1: Yes

Reviewer #2: Yes

5. Is the manuscript presented in an intelligible fashion and written in standard English?

Reviewer #1: Yes

Reviewer #2: Yes

6. Review Comments to the Author

Reviewer #1: (No Response)

Reviewer #2: Many thanks for addressing the issues raised on initial review. I think supplemental figure 2 would be better if included in the body of the manuscript. That aside, no other recommendations.

7. PLOS authors have the option to publish the peer review history of their article (what does this mean?). If published, this will include your full peer review and any attached files.

Reviewer #1: **Yes: **Philip A. Verhoef, MD, PhD

Reviewer #2: No

---

## [Author Response · Author response to Decision Letter 1]

14 Dec 2020

Dear PLOS ONE Academic Editor and Team

Thank you for your carefully considered review, detailing your helpful suggestions. I will address each point individually below, in red. 

1. The delivery of CPAP in your study needs to be clarified under many aspects. 1) Which interface was used? In the discussion, you state that 14 patients used helmet, while (I assume) the large part used a tight face mask. Report the exact numbers in the methods section and, if the sentence is correct, reword in the abstract “Sixty-nine patients were trialled on CPAP, mostly delivered by face mask, …”. 2) Which type of system was used to provide flow? Is it a free-flow Venturi, (I guess that’s not the case, since you state “CPAP is used in the UK in preference to High Flow Nasal Oxygen, largely due to concerns about oxygen supplies” and you refer to 10-15 l/min of oxygen consumption) a Bussignac valve system, a ventilator/turbine system? I guess that you used different systems for helmets and masks. Please add all the requested information in the methods section.

Thank you. The Procedures section of Methods has been amended to address the above points, and now reads as follows:

“In this study, patients not maintaining oxygen saturations over 92- 94% on 40 - 60% oxygen via a Venturi mask were commenced on CPAP 10 cm H20 and 10 litres oxygen, adjusted according to physician discretion. CPAP on the ward was delivered by the Breas Medical NIPPY 3+© ventilator, with oxygen entrained from the wall via piped oxygen attached to a flow meter. Pressure could be adjusted as required, to a maximum of 15 cm H2O and flow could be adjusted to a maximum of 15 litres O2. The default interface used on the ward was a full face mask, but a total face mask was used in a small number of patients who could not tolerate this.

With only one exception, CPAP as a ceiling of care was started on the respiratory wards in all cases. CPAP as a bridge to IMV was started on the respiratory wards in most cases. These wards consisted of three bays of four beds, and four side-rooms consisting of one bed. All beds had access to a wall-mounted oxygen supply and could support the use of CPAP. One nurse would typically look after four to eight patients. 

Oxygen requirements, which were administered as the minimum required to maintain target oxygen saturations within the range set by national guidelines, were documented as a proxy marker for hypoxemia and hence for severity of disease.

Fourteen patients requiring CPAP were commenced on CPAP Hoods instead of face masks and this was always delivered on ITU, either via the Hamilton-S1©, or the Hamilton-C3© Ventilator. With one exception, all the CPAP Hood patients remained for full escalation. “

- abstract: there are conflicting information: “Sixty-nine patients were trialled on CPAP either as a ceiling of care (N=25) or as a potential bridge to IMV (N=44).” but later you state “Among all patients trialled on CPAP either as a potential bridge to IMV (N=44) or as a ceiling-of-care (N=19) mortality...”. please report only one of the two sentences to avoid confusion (is it 25+44 or 19+44?) and check consistency with figure 2 of the supplements

Thank you. 

We have now updated the findings section of the abstract to read “Sixty-nine patients were trialed on CPAP, mostly delivered by face mask, either as an early ceiling of care instituted within 24 hours of admission (N=19), or as a potential bridge to IMV (N=44). Patients receiving a ceiling of care more than 24 hours after admission (N=6) were excluded from the analysis.”

The results section reads:

“When all patients who ever received CPAP (including those who went on to require IMV N=69) are examined, 19 patients had an early ceiling-of-care plan…”

In the Discussion section, we have amended the opening sentence to read:

“In our cohort of 294 hospitalized patients, 69 were trialed on CPAP either as a ceiling-of-care (N=25), of which only those with an early ceiling-of-care (N=19) were included in the analysis, or as a potential bridge to IMV (N=44).”

We hope this clarifies this point, and achieves better consistency for readers. The six patients unaccounted for in the above were patients who had a ceiling of care introduced later during the admission. As described in lines 174- 176 of the Statistical analysis “Patients who had a ceiling-of-care instituted more than 24h after admission were omitted as reverse causality was possible, i.e. that a ceiling-of-care was introduced following failure to respond to treatment.” 

- abstract: please report some data about use of CPAP in patients without ceiling of care. Add a sentence like “Among patients without ceiling of care (n=AA), a CPAP trial was started in BB, while CC needed prompt intubation; CPAP failure leading to intubation occurred in DD, while EE patients successfully recovered without intubation maximally receiving CPAP”

If I desumed the numbers correctly, it should be: AA=156; BB=44; CC=10; DD about 26; EE=18

Thank you. 

I think AA=156 includes the 19 patients whose ceiling of care permitted CPAP but not IMV. The corrected number, for patients without a ceiling of care, is therefore 137 (see Table 1). The final numbers should therefore be:

AA = 137; BB=44; CC=10; DD= 25, EE= 19

I have added the following text to the abstract:

“Among patients without a ceiling of care (N= 137), 10 patients required prompt intubation following failed oxygen therapy, but 44 patients received CPAP. CPAP failure, defined as death (N=1) or intubation (N=24), occurred in 57% (N= 25) of patients. But in total, 75% (N=33) of those started on CPAP with no ceiling of care recovered to discharge - 19 without the need for IMV, and 14 following IMV.”

This corresponds to the Mortality by respiratory support received and ceiling-of-care subsection of the Results section, where is states that “Among those without any ceiling-of-care (N=44), 75% recovered and were discharged – 43.1% (N=19) without requiring IMV and 31.8% (N=14) after subsequently receiving IMV (Figure-2). “

- Figure 1: add the raw number of patients for each column

This has now been done. For consistency, I have also added another version of Figure 2, depicting the raw numbers, but left the original one as well, in the hope that the editorial board may decide which version they prefer. The authors’ preference would be with the second version depicting the raw numbers. 

- Figure 2 supplements: add the number of excluded patients (apparently numbers are missing)

This has now been done. 

Once again, thank you very much for your thorough review of the manuscript, which we have all found very helpful and instructive.

Kind regards,

Jonathan Walker

---

## [Editor Report · Decision Letter 2]

18 Dec 2020

The role of CPAP as a potential bridge to invasive ventilation and as a ceiling-of-care for patients hospitalised with Covid-19 - an observational study.

PONE-D-20-20968R2

Dear Dr. Walker,

We’re pleased to inform you that your manuscript has been judged scientifically suitable for publication and will be formally accepted for publication once it meets all outstanding technical requirements.

Kind regards,

Andrea Coppadoro

Academic Editor

PLOS ONE

Additional Editor Comments (optional):

In your final submission you can include the revised version of figure 2 (the numbered version); please change the font color on the blue bars (black doesn't read).

---

## [Editor Report · Acceptance letter]

22 Dec 2020

PONE-D-20-20968R2 

The role of CPAP as a potential bridge to invasive ventilation and as a ceiling-of-care for patients hospitalized with Covid-19 - an observational study. 

Dear Dr. Walker:

I'm pleased to inform you that your manuscript has been deemed suitable for publication in PLOS ONE. Congratulations! Your manuscript is now with our production department. 

Kind regards, 

on behalf of

Dr. Andrea Coppadoro 

Academic Editor

PLOS ONE